# Effect of the Proximity to the Quintero-Puchuncaví Industrial Zone on Compounds Isolated from *Baccharis macraei* Hook. & Arn: Their Antioxidant and Cytotoxic Activity

**DOI:** 10.3390/ijms25115993

**Published:** 2024-05-30

**Authors:** Manuel Martínez-Lobos, Estela Tapia-Venegas, Paula Celis-Plá, Joan Villena, Carlos Jara-Gutiérrez, Alexandra Lobos-Pessini, Daniela Rigano, Carmina Sirignano, Alejandro Madrid-Villegas

**Affiliations:** 1Programa de Doctorado Interdisciplinario en Ciencias Ambientales, Facultad de Ciencias Naturales y Exactas, Universidad de Playa Ancha, Valparaíso 2360004, Chile; 2Laboratorio de Productos Naturales y Síntesis Orgánica, Universidad de Playa Ancha, Av. Leopoldo Carvallo 270, Valparaíso 2360004, Chile; 3Departamento de Ciencias Naturales y Geografía, Facultad de Ciencias Naturales y Exactas, Universidad de Playa Ancha, Valparaíso 2360004, Chile; 4Departamento de Ciencias de la Ingeniería para la Sostenibilidad, Facultad de Ingeniería, Universidad de Playa Ancha, Valparaíso 2360004, Chile; estela.tapia@upla.cl; 5Laboratorio de Bioprocesos, HUB Ambiental, Universidad de Playa Ancha, Valparaíso 2360004, Chile; 6Laboratorio de Investigación Ambiental Acuática (LACER), HUB Ambiental, Universidad de Playa Ancha, Valparaíso 2360004, Chile; 7Centro Interdisciplinario de Investigación Biomédica e Ingeniería Para la Salud (MEDING), Escuela de Medicina, Facultad de Medicina, Universidad de Valparaíso, Valparaíso 2540064, Chile; 8Centro Interdisciplinario de Investigación Biomédica e Ingeniería Para la Salud (MEDING), Escuela de Kinesiología, Facultad de Medicina, Universidad de Valparaíso, Valparaíso 2540064, Chile; 9Laboratorio Silob Chile, Javiera Carrera 839, Valparaíso 2390410, Chile; 10Department of Pharmacy, School of Medicine and Surgery, University of Naples Federico II, Via Montesano 49, 80131 Naples, Italy

**Keywords:** heavy metals, abiotic stress, secondary metabolites, medicinal properties

## Abstract

*Baccharis macraei* Hook. & Arn (Asteraceae), commonly known as Vautro, is found in the coastal areas of central-southern Chile, including the industrial zone of Quintero-Puchuncaví, known for the contamination of its soils with heavy metals, which together with other factors generate abiotic stress in plant species, against which they present defensive mechanisms. For this reason, the objective was to evaluate the effect of abiotic stress generated by the proximity of *B. macraei* to the industrial complex by assessing the physiological and metabolic states reported by the extracts and compounds isolated from the species, as well as the photosynthetic capacity, metal content and production, and antioxidant activity and cytotoxicity against tumorigenic cell lines of the phytoconstituents. To this end, *B. macraei* was collected at two different distances from the industrial complex, observing that the closer the species is, the greater the concentration of copper in the soil, generating a decrease in the rate of electron transport in situ, but an increase in antioxidant activity with low cytotoxicity. This activity could be due to the presence of flavonoids such as Hispidulin, Cirsimaritina, and Isokaempferida, as well as monoterpenes, oxygenated and non-oxygenated sesquiterpenes identified in this study.

## 1. Introduction

The Asteraceae family is one of the most diverse and numerous plant families on the planet, due to the historical presence it has, which places it in the southern cone in the late stage of the Cretaceous period [1,2,3]. This family stands out for its large number of genera and species and wide distribution. Therefore, there is a high diversity of habitats [4,5,6] that comprises nearly 1700 genera [7], with more than 25,000 species [1,8,9]. Its members are widely dispersed throughout the world and are found in most climatic conditions and habitats, including high-altitude forests and grasslands, with exceptions only at the poles, such as the Antarctic Continent [7,9,10] and the Greenland hinterland [11]. Numerous species of Asteraceae have medicinal applications and have been implemented in the pharmaceutical, cosmetic, and food industries [12,13]. Furthermore, due to their high content of flavonoids and phenolic acids, these species show strong antioxidant activity [14,15].

The genus *Baccharis* belongs to the Asteraceae family and comprises around 500 species distributed in North and South America, especially in the warm temperate and tropical zones of Brazil, Argentina, Colombia, Chile, and Mexico [16]. Antioxidant, anti-inflammatory, and gastroprotective activities have been described for this genus [17]. Its wide use in popular medicine in South America stands out as a source of anti-inflammatory and gastric-protective agents [18]. According to the literature, more than 100 species of *Baccharis* have been investigated to reveal that they contain many kinds of secondary metabolites. The most widespread compounds are the diterpenoids clerodane and labdane, as well as the triterpenoids of the oleanane series [19]. In Chile, there are 48 species of *Baccharis* [20], of which *Baccharis macraei* is a common exponent in the fifth region of Chile, found abundantly in the coastal communes of Valparaíso, such as Quintero and Puchuncaví. Therefore, they would be exposed to abiotic stress.

The Puchuncaví Valley, an agricultural area located 35 km north of the city of Valparaíso (V Region of Chile), has been persistently impacted for the last five decades by atmospheric pollutants and the atmospheric deposition of particles rich in metals from the various activities within a “Las Ventanas” industrial complex that includes coal-fired power plants, a copper refinery and smelter, natural gas terminals, cement companies, and other types of industries [21,22,23]. The effects of these depositions with historical trace elements remain latent in the soil of this area despite current environmental regulations. The background concentrations of Cu, As, Pb, and Zn are close to 100, 16, 35, and 122 mg/kg, respectively [24,25]. In particular, the areas of La Greda and Los Maitenes present greater soil contamination due to their proximity to the industrial complex and the predominant southwesterly winds [21,26].

The accumulation and over-enrichment of heavy metals due to industrialization pose a threat to global vegetation [27]. Plants tend to accumulate contaminating elements due to their great capacity to adapt to the chemical variabilities of the environment [28]. Regarding their defensive responses due to abiotic stress by toxic heavy metals, the plants themselves have some resilience, mainly through HMS-induced ROS scavenging, chelation through metallothionein and/or phytochelatin, compartmentalization in cell walls and intracellular vesicles, and inhibition of direct soil absorption [27].

The objective of this study was to evaluate the effect of abiotic stress generated by the vicinity of a contaminated area resulting from industrial activity on the endemic species *Baccharis macraei* Hook. & Arn by evaluating the health status of the species under study through their photosynthetic capacity and the absorbed metal content. In addition, the production and antioxidant activity of the phytoconstituents, as well as the cytotoxic capacity of the extracts and the phytoconstituents that were isolated from them, were analyzed.

## 2. Results

### 2.1. Metals Contained

As presented in Table 1, the Near Zone rhizosphere contained significant concentrations of copper and zinc (44.8 and 48.2 mg kg^−1^, respectively), whereas the Far Zone contained lower concentrations of these metals (29.45 and 37.3 mg kg^−1^, respectively). This contrasted with the leaves of the Near Zone species compared to the Far Zone species, which had lower contents of copper (11.4 mg kg^−1^ versus 15.1 mg kg^−1^) and zinc (18.2 and 18.35 mg kg^−1^), respectively.

### 2.2. Photosynthetic Capacity of Collected Plants

The physiological behavior of Near Zone and Far Zone species of *B. macraei* was assessed through measurements of their photosynthetic capacity using functional indicators such as in situ photosynthetic capacity (Yield II), maximum quantum yield (F_v_/F_m_) (chlorophyll a fluorescence associated with Photosystem II), and in situ electron transport rate (ETR in situ). All values were statistically significant (*p* < 0.05) except F_v_/F_m_. According to Figure 1, Yield II was higher in the Near Zone (0.711 ± 0.010) than in the Far Zone (0.679 ± 0.018), as was the F_v_/F_m_ ratio (0.772 ± 0.016 and 0.764 ± 0.015, respectively), while ETR in situ was higher in the Far Zone than in the Near Zone (353.67 ± 9.37 and 261.32 ± 3.76 µmol m^−2^ s^−1^, respectively).

### 2.3. Determination of Phytoconstituents from Extract

The phytoconstituents of the extract were measured using ethyl acetate, and the percentage extraction yield of the ethyl acetate extract for the *B. macraei* sample was 10.175% for the Near Zone and 7.539% for the Far Zone. Table 2 shows the phytoconstituent content of the ethyl acetate leaf extracts of the species from both study areas, showing the concentrations of the phytoconstituents in studies obtained by UV–visible spectrophotometry, which allows determining the concentration of the phytoconstituents. Analyzed in solution, the tests carried out are Phenols, Flavonoids, and Anthraquinone. These data underwent statistical analysis, with significant differences observed in the phenol and flavonoid content of the species from the Near Zone and the species from the Far Zone, with *p* < 0.05. However, no significant difference was found in the anthraquinone content (*p* > 0.05), indicating similarity in this type of phytoconstituent between the two zones. It is noteworthy that for flavonoids, the Far Zone has a higher phenol content than the Near Zone (about 30% higher), unlike flavonoids, where it is higher for the Near Zone (about 30% lower).

### 2.4. Antioxidant Activity from Extract

To assess antioxidant activity, spectrophotometric techniques were employed to conduct DPPH, FRAP, and TRAP tests. The results are displayed in Table 3. It can be observed that in most cases, there are differences between the extracts of species from the Near Zone and those from the Far Zone. Notably, the statistical analysis for the DPPH test yielded a *p*-value > 0.05, indicating that the results of this test are not significantly different between the two zones. Conversely, the Far Zone exhibits higher antioxidant activity according to the FRAP test, whereas the species from the Near Zone display greater antioxidant activity in the TRAP test. When comparing the antioxidant activity of the extract under study with the positive controls TROLOX^TM^, Gallic Acid (GA), and Butylhydroxyanisole (BHA), all the results show significant differences, with *p* < 0.05. However, the DPPH activity is notably close to that of the controls used.

### 2.5. Identification of Phytoconstituents from Extracts Using GC-MS 

Volatile and semi-volatile chemical compounds in the ethyl acetate extract of *B. macraei* from the Near and Far Zones were analyzed using GC-MS. The GC-MS chromatograms are depicted in Appendix A, and Table 4 displays the relative area of the detected compounds in the extract. A total of 13 compounds were observed in the Far Zone sample, while the Near Zone sample showed 15 compounds. Among these compounds, five are common to both extracts: 2-Pentanone, 4-hydroxy-4-methyl- (**1**); α-pinene (**2**); β-pinene (**3**); (-)-Spathulenol (**4**), and Cis-β-Farnesene (**5**) (Figure 2). Additionally, it is important to note the increase in the relative area of these compounds in the Near Zone sample compared to the Far Zone sample. Furthermore, the presence of sesquiterpene compounds is highlighted in the Near Zone extract and not in the Far Zone extract.

According to Figure 3A, the number of compounds per chemical class in the Near Zone sample demonstrates an increased diversity of sesquiterpenes and monoterpenes compared to the Far Zone. In addition, in Figure 3B, the total area occupied by sesquiterpenes, oxygenated sesquiterpenes, monoterpenes, diterpenes, and aromatic compounds accounts for 33.33%, 16.67%, 33.33%, 0.00%, and 16.7% in the Far Zone, respectively. In contrast, in the Near Zone sample, these percentages are 33.33%, 16.67%, 41.67%, 8.33%, and 0.00%, respectively.

### 2.6. Identification of Compounds Isolated from the Extract

Through the isolation process, five fractions were isolated from both samples. Among them, two fractions correspond to the samples from the Near Zone of *B. macraei* and three fractions correspond to the ethyl acetate extract of the sample from the far zone of *B. macraei*. Among the group of fractions, compounds 4, 9, 12, 13, and 14 have already been cited in the literature, and their spectra can be found in Appendix A. The ^1^H and ^13^C peaks are described below.

Hispidulin (**6**): Fractions 14 and 9 of the Far Zone sample and the Near Zone sample, respectively, turned out to be a fine opaque yellow powder. It was dissolved in deuterated methanol (MeOD) and subjected to ^1^H-NMR (400 MHz, MeOD) and ^13^C-NMR (100.1 MHz, MeOD) spectroscopy to determine its structure, identifying the compound Hispidulin. In this way, the ^1^H NMR spectrum resulted as follows: *δ* 7.85 (*d*, *J* = 20.01 Hz. 1H, H-2′ and H-6′); 6.92 (*d*, *J* = 8.00 Hz, 1H, H-3′ and H-5′); 6.60 (*s*, 1H, H-3); 6.57 (*s*, 1H, H-8); 3.9 (*s*, 1H, OH); 3.88 (*s*, 3H, OCH_3_); 3.75 (*s*, 1H, OH). Regarding the ^13^C-NMR spectrum, the peaks are the following: δ 184 (C-4); 165 (C-2); 161 (C-4′); 159 (C-7); 154 (C-9); 153 (C-5); 131 (C-6); 129 (C-2′ and C-6′); 122 (C-1′); 115 (C-3′ and C5′); 105 (C-10); 103 (C-3); 95 (C-8); 60 (O-Me).

Mixture of Hispidulin (**6**) and Cirsimaritine (**7**): Fraction 13 corresponding to the Far Zone turned out to be a yellow powder that represents a mixture of two flavonoids. This mixture was dissolved in deuterated methanol for ^1^H NMR testing (400 MHz, MeOD) and ^13^C NMR (100.1 MHz, MeOD) spectroscopy to determine its structure. The compounds in this mixture were identified and compared with the literature, identifying Hispidulin and Circimaritine. The ^1^H NMR peaks for Hispidulin were identified as: δ 7.85 (*d*, *J* = 20.01 Hz. 1H, H-2′ and H-6′); 6.92 (*d*, *J* = 8.00, 1H, H-3′ and H-5′); 6.60 (*s*, 1H, H-3); 6.57 (*s*, 1H, H-8); 3.9 (*s*, 1H, OH); 3.88 (*s*, 3H, OCH_3_); 3.75 (*s*, 1H, OH). Regarding the ^13^C-NMR spectrum, the peaks are the following: δ 184 (C-4); 165 (C-2); 161 (C-4′); 159 (C-7); 154 (C-9); 153 (C-5); 131 (C-6); 129 (C-2′ and C-6′); 122 (C-1′); 115 (C-3′ and C5′); 105 (C-10); 103 (C-3); 95 (C-8); 60 (O-Me). On the other hand, the peaks that identify the compound Cirsimaritine in the ^1^H NMR spectrum correspond to the following: δ 7.91 (*d*, *J* = 4.0013 Hz. 1H, H-2′ and H-6′); 6.95 (*d*, *J* = 4.0013 Hz. 1H, H-8); 6.94 (*d*, *J* = 4.0013 Hz. 1H, H-3′ and H-5′); 6.84 (*s*, 1H, H-3); 3.99 (*s*, 3H, 6-OMe); 3.84 (*s,* 3H, 7-OMe), while for the ^13^C NMR spectrum, the peaks are the following: δ 183 (C-4); 165 (C-2); 162 (C-4′); 159 (C-7 and C-9); 150 (C-5); 132 (C-6); 128 (C-2′ and C-6′); 122 (C-1′); 115 (C-3′, C-5′ and C-8); 105 (C-10); (C-4′); 91 (C-3); 60 (6-OMe); 55 (7-OMe).

Mixture of Hispidulin (**6**) and Isokaempferide (**8**): Fraction 12 of Far Zone turned out to be a yellow powder that represents a mixture of two flavonoids. This mixture was dissolved in deuterated methanol for testing by ^1^H NMR (400 MHz, MeOD) and ^13^C NMR (100.1 MHz, MeOD) spectroscopy to determine its structure. The compounds in this mixture were identified and compared with the literature, identifying Hispidulin and Isokaempferide. The ^1^H NMR peaks for Hispidulin were identified as: δ 7.85 (*d*, *J* = 20.01 Hz. 1H, H-2′ and H-6′); 6.92 (*d*, *J* = 8.00, 1H, H-3′ and H-5′); 6.60 (*s*, 1H, H-3); 6.57 (*s*, 1H, H-8); 3.9 (*s*, 1H, OH); 3.88 (*s*, 3H, OCH_3_); 3.75 (*s*, 1H, OH). Regarding the ^13^C-NMR spectrum, the peaks are the following: δ 184 (C-4); 165 (C-2); 161 (C-4′); 159 (C-7); 154 (C-9); 153 (C-5); 131 (C-6); 129 (C-2′ and C-6′); 122 (C-1′); 115 (C-3′ and C5′); 105 (C-10); 103 (C-3); 95 (C-8); 60 (O-Me). With respect to the ^1^H NMR of Isokaempferide, the peaks are the following: 8.00 (*d*, *J* = 8.00 Hz. 1H, H-2′ and H-6′); 6.93 (*d*, *J* = 8.00 Hz, 1H, H-3′ and H-5′); 6.41 (*s,* 1H, H-8); 6.21 (*d*, *J* = 4.0013 Hz); 3.78 (*s*, 3H, 3-OMe). Regarding the peaks determined by ^13^C-NMR, they are the following: 178 (C-4); 165 (C-5); 161 (C-9); 160 (C-4′); 157 (C-7); 155 (C-2); 138 (C-3); 130 (C-2′ and 6′); 121 (C-1′); 115 (C-3′ and 5′); 105 (C-10); 98 (C-6); 93 (C-8); 58 (C-3 OMe).

Hautriwaic acid (**9**): Fraction 4 of the Near Zone corresponds to needle-shaped crystals that were dissolved in deuterated methanol (MeOD) and subjected to ^1^H NMR (400 MHz, MeOD), yielding the following peaks that identify it as hautriwaic acid: δ 7.38 (*t*, *J* = 4.0013 Hz. 1H, H-15); 7.26 (*s*, 1H, H-16); 6.68 (*t*, 1H, H-3); 6.29 (*s*, 1H, H-14); 4.14 (*d*, *J* = 8.0026 Hz.1H, H-19); 3.75 (*d*, *J* = 12.0039 Hz.1H, H-19); 2.28 (*m*, *J* = 28.0091 Hz. 2H, H-1); 2.28 (*m*, *J* = 32.0104 Hz. 2H, H-2); 1.71 (*m*, *J* = 4.0013 Hz. 2H, H-7); 1,68 (*m*, *J* = 4.0013 Hz. 2H, H-8 and H11); 1.55 (*s*, 2H, H-12), 1.47 (*s*, 2H, H-6), 1.13 (*m*, *J* = 16.0052 Hz.1H, H-10); 0.88 (*d*, *J* = 8.0026 Hz.3H, H-17), 0.80 (*s*, 3H, H-20). ^13^C-NMR (100 MHz, MeOD): δ 172.1 (C-18); 142.0 (C-15); 140 (C-4); 138.0 (C-16); 138.0 (C-3); 125 (C-13); 110.5 (C-14); 64.5 (C-19); 46 (C-10); 41.5 (C-9); 38 (C-11); 38 (C-5); 36 (C-8); 31 (C-6); 26 (C-2); 26 (C-7); 18.1 (C-20); 18 (C-12); 16.5 (C-1); 15.3 (C-17).

### 2.7. Cytotoxicity Activity of B. macraei Extracts and Isolated Compounds

The cytotoxicity of the extracts from *B. macraei* species from both the Far and Near Zones, as well as of the isolated compounds from both samples, was evaluated in three tumorigenic cell lines corresponding to PC-3, HT-29, and MCF-7, as well as in a non-tumorigenic cell line corresponding to MCF-10. This evaluation was conducted using the Sulforhodamine-B assay, which provides the percentage of cells that have survived treatment with the extracts. These results are presented in Table 5 and Table 6, where it can be observed that for all treatments, regardless of the cell lines used, the origin of the extracts (Near Zone or Far Zone), or the different isolated compounds, high cytotoxicity was not observed, with IC_50_ values of more than 100 µg mL^−1^ obtained.

### 2.8. Correlation between the Environmental Variables Measured in the Extract

Table 7 and Table 8 present the Pearson correlation coefficient between the content of phytoconstituents (measured in terms of phenols and flavonoids) and the antioxidant activity (measured by the DPPH, FRAP, and TRAP assays) of the extracts under study for both extracts studied (from Far and Near Zones). Additionally, it shows the correlation of these assays with the results of photosynthetic activity represented by the in situ ETR (Yield II was not considered since it is directly proportional to the in situ ETR according to Equation (1)), as well as the copper levels measured in the leaves and rhizospheres of the plants. For all variables, Pearson correlation values closer to 1 or −1 indicate a higher degree of positive or negative correspondence, respectively (only values above 0.8 or below −0.8 are considered significant in this study). It is observed that the correlations between the species of the Near Zone and the Far Zone show several differences, indicating distinct behaviors.

For the Far Zone (see Table 7), the results indicate a strong negative correlation between phenols and copper in the rhizosphere (r > −0.9 and *p* < 0.05). Similarly, a high negative correlation is observed between flavonoids, antioxidant activity measured by FRAP and TRAP (r > −0.9 and *p* < 0.05), copper in the rhizosphere (r > −0.9 and *p* < 0.05), and in situ ETR (r > −0.9 and *p* < 0.05). Additionally, antioxidant activity measured by FRAP and TRAP and in situ ETR are positively correlated with copper in the plant’s rhizosphere (r > 0.8 and *p* < 0.05).

For the Near Zone (see Table 8), the results indicate a strong positive correlation between phenols, antioxidant activity measured by DPPH and FRAP (r > 0.8 and *p* < 0.05), and in situ ETR (r > 0.9 and *p* < 0.05), while a negative correlation is observed regarding antioxidant activity measured by TRAP (r > −0.9 and *p* < 0.05). This implies that an increase in phenol content will lead to higher antioxidant activity in both DPPH and FRAP assays, as well as an increase in the electron transport rate in situ (ETR in situ). Additionally, a strong positive correlation is observed between flavonoids and copper in the rhizosphere (r > 0.9 and *p* < 0.05) and a negative correlation with copper in leaves (r > −0.9 and *p* < 0.05). Furthermore, in situ ETR is positively correlated with antioxidant activity measured by DPPH and FRAP (r > 0.8 and *p* < 0.05) and negatively correlated with antioxidant activity measured by TRAP (r > −0.9 and *p* < 0.05). Finally, it is important to highlight the negative correlation between the copper content in the leaves and the copper content in the rhizospheres of the species (r = −1 and *p* < 0.5).

## 3. Discussion

### 3.1. Metals Contained

In relation to the metal content in the rhizospheres in both study areas, the results presented lower contents than those reported in areas close to the industrial complex such as La Greda and Los Maitenes, where metals and metalloids such as Cu, As, Pb, and Zn have concentrations close to 100, 16, 35, and 122 mg kg^−1^, respectively [24,25]. However, in both zones studied (the Near and Far Zones), the content of Cu and Zn was higher than in the Near Zone.

Between both areas studied, the fact that the Near Zone has a higher concentration of copper compared to the Far Zone (around 50% more) is due to the greater proximity in kilometers to the industrial complex and the winds coming from the southwest with a northeasterly direction, which is justified as the Far Zone, located in the south of the complex, is less contaminated [21,26,29]. The soil contamination by heavy metals is linked to plant pollution, as the metals accumulate in the soil’s surface layer and can enter the food chain [30].

In relation to the copper content found in the leaves, the Far Zone had a higher concentration than the Near Zone (around 30% more). Plants of the Asteraceae family have already been reported as producers of metal-chelating compounds, which identifies them as plants with the capacity for bioaccumulating metals. Therefore, it is not strange that these species are capable of accumulating metals [15]. However, the content of bioaccumulated metals depends not only on the plant but also on the chemical form in which they are present in the soil, as well as the soil’s physical and chemical properties (pH, redox condition, electrical conductivity, organic matter content, and particle size composition) [30].

### 3.2. Photosynthetic Capacity

The impact of proximity to an industrial zone on the photosynthetic capacity of *B. macraei* was investigated. Functional indicators considered in assessing photosynthetic capacity included the effective quantum yield (Yield II) and the in situ electron transport rate (in situ ETR). It was observed that the Near Zone had a higher Yield II compared to the Far Zone. However, specimens located in the Near Zone showed lower in situ ETR values compared to those in the Far Zone, attributed to differences in the measured incident PAR irradiance values for each specimen. It is well-established that various abiotic stresses influence the mechanism of photosynthesis [31]. Therefore, the observed differences between the Far and Near Zones would suggest variations in the extent of abiotic stress experienced by each studied plant, with the Near Zone being subjected to higher levels of stress.

### 3.3. Determination of Phytoconstituents from Extracts

There is a higher presence of phenolic compounds in the Far Zone compared to the Near Zone, unlike the concentration of flavonoids, where the Near Zone presents a higher content than the Far Zone. On the other hand, the content of anthraquinones is very low in both types of samples. These differences between the Near and Far Zones can be understood because plant species synthesize the content of these phytoconstituents depending on their responses to environmental factors to which they are being exposed. For example, it has been indicated that abiotic stress, such as the presence of metals, could increase the biosynthesis of some phytoconstituents [32].

The presence of flavonoids in *Baccharis* is expected since it has been described that they are common secondary metabolites in this genus, and phenols are the largest group of secondary metabolites in plants in general and are observed in plants used for therapeutic purposes [19].

### 3.4. Antioxidant Activity from Extracts

The Near Zone sample showed an increase in IC_50_ value compared to the Far Zone sample (around 13% higher), indicating higher antioxidant activity in the Far Zone using the DPPH method. The same trend is observed when measuring activities using the FRAP assay; the Far Zone extract showed higher antioxidant activity than the Near Zone, with an increase of nearly 20%. However, using the TRAP assay, the Far Zone has lower antioxidant activity than the Near Zone. The extracts from both areas had a value 700 to 800% higher than the antioxidant used as the positive control, Trolox^TM^, in the DPPH test, indicating lower antioxidant activity compared to the positive control. However, for the FRAP and TRAP tests, both samples showed between 70 to 90% less antioxidant activity than the positive controls. However, it is possible to improve the quality and quantitative characteristics of plant extracts by employing a highly precise and accurate method [33].

### 3.5. Identification of Compounds in the Extracts

Five common compounds were identified between both plants studied in the different zones, with an increase in the Near Zone compared to the Far Zone. Additionally, the Near Zone contains more sesquiterpenes and monoterpenes than the Far Zone. The Near Zone had a higher quantity of monoterpenes than sesquiterpenes, unlike the Far Zone, which had an equal percentage of both phytoconstituents. Other studies have already evaluated other species of this genus, such as Ueno et al. in 2018 [34], where three sesquiterpenes, three diterpenes, two alkenyl p-coumarates, and one flavonoid were identified for *B. retusa*. Another study conducted by Ascari et al. in 2019 [35] found that another genus, *B. punctulata*, had a lower proportion of monoterpenes compared to sesquiterpenes, contrary to what was observed in the Far Zone.

### 3.6. Isolation and Identification of Compounds

Regarding the NMR spectrum of the five isolated compounds, which were Fraction 14 (Hispidulin (**6**)), Fraction 13 (mixture of Hispidulin (**6**) and Cirsimaritin (**7**)), Fraction 12 (mixture of Hispidulin (**6**) and Isokaempferide (**8**)), Fraction 9 (Hispidulin (**6**)), and Fraction 4 (Hautriwaic acid (**9**)), the first three were identified in the Far Zone, while the next two compounds were identified in the Near Zone extract, highlighting the presence of Hispidulin in both areas, which is a compound for the genus [36].

Hispidulin (**6**): In Fractions 9 and 14, the compound identified was hispidulin. This identification was carried out using spectroscopic techniques in which the peaks of the ^1^H and ^13^C NMR spectra were compared with those in the literature [37]. The correlation of the hydrogens with the carbon to which they are bonded was carried out using the HSQC technique. In addition, an HMBC was carried out where the typical chemical shifts of the B-aromatic ring, typical of flavonoids, are evident. In this way, the shift *δ_H_* 7.85 (H-2′) is correlated with *δ_C_* 165 (C-2), 161 (C-4′), and 115 (C-3′), which are located at distances of three, three, and two bonds, respectively ($J_{H2\prime -C2}^{3}$, $J_{H2\prime -C4\prime }^{3}$, and $J_{H2\prime -C3\prime }^{2}$). Likewise, the shift of δ_H_ 6.92 (H-3′) correlates with *δ_C_* 122 (C-2′), the shift *δ_H_* 6.92 (H-5′) correlates with *δ_C_* 129 (C-6′), and *δ_H_* 7.85 (H-6′) correlates with *δ_C_* 115 (C-5′), all two bonds away ($J_{H3\prime -C2\prime }^{2}$, $J_{H5\prime -C6\prime }^{2}$ and $J_{H6\prime -C5\prime }^{2}$). On the other hand, the displacement *δ_H_* 6.60 (H-3) is correlated with *δ_C_* 122 (C-1′) of the B ring and *δ_C_* 184 (C-4), 105 (C-10), and 103 (C-3), of the C ring, where the bond distances are three, two, three, and one, respectively ($J_{H3-C1\prime }^{3}$, $J_{H3-C4}^{2},$ $J_{H3-C10}^{3},$ and $J_{H3-C3}^{1}$). Furthermore, a correlation is observed in the displacement *δ_H_* 6.57 (H-8) with *δ_C_* 159 (C-7), 154 (C-9), 153 (C-5), 131 (C-6), and 95 (C-8), for ring A, with bond distances of two, two, four, three, and one, respectively ($J_{H8-C7}^{2}$, $J_{H8-C9}^{2},$ $J_{H8-C5}^{4}$, $J_{H8-C6}^{3},$ and $J_{H8-C8}^{1}$). Figure 4 shows the correlations described.

Mixture of Hispidulin (**6**) and Cirsimaritine (**7**): The presence of two compounds in fraction 13 of the ethyl acetate extract from *B. macraei* in the Far Zone was determined from ^1^H NMR, where the characteristic peaks of the aromatic B ring of a flavonoid (*δ*_H_ 7.91 and 6.94 for the compound cirsimaritine and *δ*_H_ 7.87 and 694 for hispidulin) can be seen. It also shows the presence of three methoxyl groups, one for hispidulin [37] and two for cirsimaritine, which are found at *δ*_H_ 3.88, 3.99, and 3.84, respectively. From this, HSQC analysis confirms the correlation of each hydrogen with the carbon to which it is bonded [38]. In the HMBC assay, the following heteronuclear interactions are observed, in which the correlation of *δ_H_* 7.91 (H-2′) with *δ_C_* 115 (C-3′), 162 (C-4′), and 165 (C-2), which have bond distances of two, three, and three bonds, respectively, with respect to H-2′ ($J_{H2\prime -C3\prime }^{2}$, $J_{H2\prime -C4\prime }^{3}$, and $J_{H2\prime -C2}^{3}$). In addition, the displacement of *δ_H_* 6.93 (H-3′) with *δ_C_* 122 (C-1′) is observed three links away ($J_{H3\prime -C1\prime }^{3}$). As for rings A and C, the classification is established mainly from H-3, where *δ_H_* 6.84 correlates with *δ_C_* 183 (C-4), 159 (C-9), and 105 (C-10). The link distances to which they are found, with respect to H-3 is, two, four, and three, respectively ($J_{H3-C4}^{2}$, $J_{H3-C9,}^{4}$ and $J_{H3-C10}^{3}$). Finally, the carbons of ring A were related to H-OMe of carbon 6 and 7, observing the following displacements: *δ_H_* 3.84 (H-6-OMe) with *δ_C_* 150 (C-5) and 132 (C-6), and *δ_H_* 3.99 (H-7-OMe) with *δ_C_* 159 (C-7), which are found at four, three, and three bonds ($J_{H6OMe-C5}^{4}$, $J_{H6OMe-C6}^{3}$, and $J_{H7OMe-C7}^{3}$). Figure 4 shows the correlations described.

Mixture of Hispidulin (**6**) and Isokaempferide (**8**): In fraction 12 of the *B. macraei* Far Zone ethyl acetate extract, two known compounds were observed, which were identified from ^1^H NMR, where the characteristic peaks of the aromatic B ring of a flavonoid were determined (δH 7.99 and 6.94 for the compound isokaempferide and δ_H_ 7.87 and 694 for hispidulin). It also shows the presence of two methoxyl groups, one for hispidulin [37] and two for isokaempferide [39], which are located at *δ_H_* 3.88 and 3.78, respectively. From this, HSQC analysis confirms the correlation of each hydrogen with the carbon to which it is bonded. In the HMBC assay the following heteronuclear interactions are observed, in which the correlation of *δ_H_* 7.99 (H-2′) with *δ_C_* 115 (C-3′), 160 (C-4′), and 155 (C-2), which are two, three, and three bonds distant, respectively, with respect to H-2’ ($J_{H2\prime -C3\prime }^{2}$, $J_{H2\prime -C4\prime }^{2},$ and $J_{H2\prime -C2}^{3}$). In addition, the displacement of *δ_H_* 6.94 (H-3′) with *δ_C_* 130 (C-2′) and 121 (C-1′) is observed, which are two and three links away ($J_{H3\prime -C2\prime }^{2}$ and $J_{H3\prime -C1\prime }^{3}$). Continuing with ring B, the chemical shift *δ_H_* 6.94 (H-5′) is correlated with *δ_C_* 130 (C-6′), while the shift *δ_H_* 7.99 (H-6′) is otherwise correlated. Figure 4 shows the correlations described. 

Hautriwaic acid (**9**): The identification of Hautriwaic acid (Fraction 4) was carried out by means of spectroscopic techniques where the peaks of both the ^1^H and ^13^C NMR spectra were compared with the literature [40]. In addition, an HMBC was carried out where the correlation of *δ_H_* 3.68 (H-19) with *δ_C_* 140 (C-4) and 31 (C-6), both with J^3^ displacements, is evident. In addition, correlation is observed in the displacement of *δ_H_* 0.88 (H-17) with *δ_C_* 38 (C-11), 36 (C-8), and 26 (C-7), at J^4^, J^2^, and J^3^, respectively. Correlation is observed in the displacement of *δ*_H_ 6.93 (H-3) with *δ*_C_ 23.3 (C-2) and 11.5 (C-1), in J^2^ and J^3^, respectively. Furthermore, correlation with *δ_H_* 1.13 (H-10) is obtained with *δ_C_* 140 (C-4), 65 (C-19), 41.5 (C-9), 36 (C-8), and 26 (C-7), to J^3^, J^3^, J^2^, J^3^, and J^4^, respectively. Finally, correlation to *δ_H_* 6.29 (H-14) is expressed with *δ_C_* 142 (C-15), 138 (C-16), 124 (C-13), and 18 (C-12), at J^2^, J^3^, J^2^, and J^3^, respectively. From the HSQC perspective, it is observed that a correlation is established between *δ_H_* 7.38 (H-15)/*δ_C_* 142 (C-15), *δ_H_* 7.26 (C-16)/*δ_C_* 138 (C-16), *δ_H_* 6. 68 (H-3), *δ_C_* 138 (C-3), *δ_H_* 6.29 (H-14)/*δ_C_* 110 (C-14), *δ_H_* 4.14 and 3.75 (H-19)/*δ_C_* 65 (C-19), *δ_H_* 2.28 (H-2)/*δ_C_* 26 (C-2), *δ_H_* 2.24 (H-1)/*δ_C_* 16.5 (C-1), and *δ_H_* 1.71 (H-7)/*δ_C_* 26 (C-7). Figure 4 shows the correlations described.

### 3.7. Correlations among the Variables Studied

Although the Near Zone would experience greater abiotic stress conditions than the Far Zone, considering the amount of metals in the rhizosphere and the photosynthetic capacity, a direct positive relationship between copper content (in the rhizosphere or leaf) and in situ ETR is not observed according to Pearson correlation. In contrast, in the Far Zone, there is a correlation between increased in situ ETR and metal content in the rhizosphere. Previous studies have demonstrated that the presence of metals can negatively affect photosynthetic capacity because heavy metal ions can replace essential metallic elements for the functioning of certain enzymes. This alteration can disrupt protein structure and functionality, directly impacting photosynthetic capacity [41].

Related to the Far Zone, given the positive correlation observed between FRAP and TRAP antioxidant activity and copper content in the rhizosphere and ETR in situ, it could be inferred that these variables would be increasing the antioxidant activity of the species under study. The degree of copper stress in the rhizosphere would increase ETR in situ, and all of this would be conducive to enhancing its antioxidant activity, potentially even reducing its stress level. It has been described that some metals eliminate generated ROS species, helping to decrease their stress [42]. For the Near Zone, in situ ETR would be positively related to antioxidant activity measured by FRAP and DPPH. Therefore, greater in situ ETR would increase antioxidant activity.

In the Near Zone, Pearson correlation analysis revealed that the antioxidant activity measured by FRAP was attributed to the phenol content in the samples. However, this correlation was not observed in the Far Zone. Additionally, no correlation with flavonoids was observed in either sample. Nevertheless, another study reported that the species *B. sphenophylla* exhibits antioxidant activity due to the presence of a group of flavonoids with a structure similar to those identified in this study for *B. macraei*. In fact, both species contain the flavonoid Hispidulin. Furthermore, it is suggested that the low antioxidant activity of the flavonoids identified in this study is attributed to their methoxylated structures, which may reduce their capacity [43]. On the other hand, a study conducted on the species *B. erioclada* suggests that the antioxidant activity exhibited by the species is primarily attributed to the oxygenated sesquiterpene compounds found and identified in the essential oil of the species. This finding implies that similar compounds found in *B. macraei*, such as (-)-Spathulenol and 4(15),5,10(14)-Germacratrien-1-ol, could be responsible for its antioxidant activity [44].

It is important to highlight the DPPH activity, as it is close to the controls, indicating good antioxidant activity. This is due to the presence of phytoconstituents found in these species, where the presence of flavonoids and terpenes (identified by ^1^H and ^13^C NMR), combined with oxygenated and non-oxygenated monoterpenes and sesquiterpenes, makes a significant contribution to this activity. In detail, it was observed that compounds identified by GC-MS, such as Cis-Beta-Farnesene, (-)-Spathulenol, Beta-pinene, and Alpha-pinene, and by NMR, the flavonoid Hispidulin, were found in the extract obtained from both Near Zone and Far Zone species, which may be related to the antioxidant activity present. These types of compounds have been identified to possess antioxidant activity [45]. When analyzing the extracts separately, it was observed that the antioxidant activity of the extract obtained from the Near Zone was possible due to the presence of monoterpenes such as γ-Terpinene, β-Thujene, and (R)-1-Methyl-5-(1)-methylvinyl)cyclohexene, sesquiterpenes such as Ylangene and γ-Amorphene Cadina-1(10),4-diene, oxygenated sesquiterpenes such as 4(15),5,10(14)-Germacratrien-1-ol, and the diterpene Neophytadiene, all identified by GC-MS, in addition to the terpene identified by NMR, Huatriwuac acid. However, in the case of the antioxidant activity of the extract obtained from the Far Zone, its activity is possible due to the presence of the sesquiterpene identified by GC-MS, Copaene, and the flavonoids identified by NMR, Cirsimaritin and Isokaempferide.

### 3.8. Cytotoxicity of Extracts and Compounds

By evaluating the cytotoxic activity of *B. macraei* extracts on the tumorigenic lines HT-29, PC-3, and MCF-7 and the non-tumorigenic line MCF-10 using the Sulforhodamine-B assay, which determines the concentration of cells that have survived treatment with the extracts under study, no cytotoxic effect was observed in the tumorigenic cell lines HT-29, PC-3, and MCF-7, nor in the non-tumorigenic line MCF-10. However, other plants within the same genus have previously been reported to possess anticancer properties. In a study by Lucas et al. in 2021 [46], the cytotoxic activity of an extract from *B. uncinella* was investigated, revealing an antiproliferative effect on tumor cells, particularly T24 cells, with a less aggressive impact on the evaluated normal cells. Furthermore, studies related to the anticancer activity of flavonoids from the *Baccharis* genus such as hispidulin have shown that this compound has anticancer activity, which is associated with different mechanisms, such as the induction of autophagy, the ability to trigger growth inhibition, apoptotic activation, cell cycle arrest, and inhibition of metastasis. Despite this, in the tumorigenic cell models studied in this study, an activity that considerably inhibits this type of cell was not observed [36].

## 4. Materials and Methods

### 4.1. Experimental Design

Figure 5 presents the experimental design carried out for this study. *B. macraei* specimens were collected at two different distances from the “Las Ventanas” industrial complex located in the industrial zone of Quintero-Puchuncaví. The metals present and the photosynthetic capacity of the plants were measured from the *B. macraei* leaf as well as from the rhizosphere (the soil that surrounds plant roots in this study). An extract was prepared using ethyl acetate and the content of phytoconstituents and antioxidant and cytotoxic activity, as well as volatile and semi-volatile compounds, were identified. Compounds that were identified were isolated from the extracts, and the cytotoxic capacity of each isolated compound was evaluated individually in tumorigenic and non-tumorigenic cell lines.

### 4.2. Plant Collection and Metal Measurement

Considering that the industrial complex is located in “Las Ventanas” (32°76′ S, 71°48′ W), samples of *B. macraei* were collected from two areas situated at different distances from the emission zone, according to the availability of the plant in the area. The first zone was close to the Industrial complex (Near Zone), and the second in a more distant zone (Far Zone) (see map, Figure 6). The Near Zone species was collected on the Quirilluca cliff, located in the commune of Puchuncaví, in the Valparaíso region, Chile (coordinates of the study site: 32.69722° S–71.45432° W), located 11 km north of the industrial zone. Conversely, the Far Zone sample was collected from La Posada del Parque, commune of Quintero, in the Valparaíso region, Chile (coordinates of the study site: 32.88186° S–71.50254° W), located 18 Kilometers south of the industrial zone. The plant samples were taxonomically identified by the curator of the Herbarium of the University of Playa Ancha, MCs. Pamela Ramirez, and the specimens were deposited in the herbarium of the University of Playa Ancha (voucher number: VALPL PLV 2449). From the collected plants, an ICP-MS Agilent 7700 (Agilent Technologies, Tokyo, Japan) was used to measure Cu, As, Pb, and Zn concentrations in their leaves and rhizosphere. Previously, sediment sample acid digestion was conducted using a MARS 6 iWave 240/50 microwave (CEM, Matthews, NC 28104, USA). The methodology followed was based on the SM 23rd Edition Method 3030 K–3125 B [47]. The analyses were performed by Laboratory Silob Chile.

### 4.3. Photosynthesis and Energy Dissipation as In Vivo Chlorophyll 

In vivo chlorophyll fluorescence by Photosystem II was determined using a portable pulse amplitude modulated fluorometer (Mini-PAM II-Walz GmbH, Effeltrich, Germany). Basal fluorescence (F_o_) and maximum fluorescence (F_m′_) were measured under light conditions to obtain the effective quantum yield (ΔF/F_m′_ or Yield II), where ΔF/F_m′_ = (F_m′_ − F_o_)/F_m′_. To determine the maximum quantum yield (F_v_/F_m´_), F_o_ and F_m_ were measured after 15 min in darkness, with F_v_ = F_m′_ − F_o_, F_o_ being the basal fluorescence of 15 min dark-adapted thalli and Fm being the maximal fluorescence after exposure to a saturation light pulse of >4000 mmol m^−2^ s^−1^, with a duration of a few seconds [48]. The electron transport rate (ETR in situ) was determined after 20 s of exposure to twelve increasing irradiances of actinic white light (halogen lamp provided by the Diving-PAM) [49]. The ETR in situ was calculated as follows [48]:ETR in situ (mmol electrons m^−2^ s^−1^) = ΔF/F_m′_ × E × A × F_II_(1)
where ΔF/F_m′_ is the effective quantum yield, being ΔF = F_m′_ − F_t_ (F_t_ is the intrinsic fluorescence of a plant incubated in light and F_m_′ is the maximal fluorescence reached after a saturation pulse is shone on a plant incubated in light), E is the incident PAR irradiance expressed in mmol photons^−2^ s^−1^ (1108 µmol photon m^−2^ s^−1^ at 13:30 pm for the Control species and 782 µmol photon m^−2^ s^−1^ at 11:30 am for contaminated species, both species on the same day), A is the thallus absorptance as the fraction of incident irradiance that is absorbed by the plant [50], and F_II_ is the fraction of chlorophyll related to PSII (400–700 nm), being 0.5 in vascular plants [51,52,53].

### 4.4. Preparation of the Ethyl Acetate Extract

The extracts of both Near and Far Zone plant materials were obtained from 300 g of dried leaves under conditions of darkness and room temperature in the Laboratory of Natural Products and Organic Synthesis (LPNSO) of the University of Playa Ancha, Valparaíso, Chile. The extracts of the species under study were prepared with technical grade Ethyl Acetate (A) solvents, using ultrasound equipment (Elma TI-H-5, Singen, Germany) with two sessions of one hour each, separated by an interval of 24 h. The percentage yield of extracts obtained from *B. macraei* was calculated using the formula given below (Equation (2)). The concentration of the macerate (extracts plus solvent) was determined using a rotary evaporator (LabTech EV400H, Hopkinton, MA, USA) with a vacuum pump (LabTech VP30, Hopkinton, MA, USA) at 45 °C. This technique consists of extracting the solvent, that is, extracting the ethyl acetate, resulting in a dark green solid in both cases. These extracts were used to measure phytoconstituent content, antioxidant activity, and anticancer activity. In addition, they were used to isolate and then characterize the major compounds. Subsequently, the samples were stored at −4 °C.% R= (Extract/(Dry mass)) × 100(2)

### 4.5. Estimation of Phytoconstituents of B. macraei Extracts

The estimation of phenol, flavonoids, and anthraquinones of the Near and Far Zone extracts of *B. macraei* were quantitatively identified using UV–visible spectrophotometric methods, as stated in the literature.

The total phenolic content of the species under study was used with the Folin–Ciocalteu colorimetric test [54,55]. The method began by adding 0.5 mL of the extract at a concentration of 1.0 mg mL^−1^ of the dry extract to 2.5 mL of Folin–Ciocalteu reagent at a concentration of 0.2 N, generating a homogeneous mixture. After 5 min, 2 mL of 7.5% aqueous sodium carbonate (Na_2_CO_3_) solution was added and the mixture was allowed to stand in the dark for 2 h at room temperature. After 2 h of rest, the absorbance was measured at 700 nm using a UV–visible spectrophotometer (Rayleigh UV-2601). Each measurement was performed in triplicate. To establish the calibration curve, the gallic acid standard was used, and the total phenolic content was calculated and expressed as mg of gallic acid equivalents (GAE) per g of dry extract. All measurements were performed in triplicate.

The total flavonoid content of the Near and Far Zone extracts of *B. macraei* was determined using the aluminum chloride (AlCl_3_) colorimetric method [54,55]. A total of 500 mL of solution of each extract under study at a concentration of 1.0 mg mL^−1^ of dry extract was added to 500 mL of a 2% AlCl_3_ solution in ethanol. After 10 min, absorbance was measured at 415 nm using a UV–visible spectrophotometer (Rayleigh UV-2601, Beijing, China). Each sample was measured in triplicate. In addition, a standard solution was prepared as a blank, consisting of 500 mL of the extract with 500 mL of ethanol without AlCl_3_. A standard calibration curve was obtained using known concentrations of a freshly prepared quercetin solution. Therefore, the total concentration of flavonoids in the extract sample was calculated and expressed as mg of quercetin equivalents (QE) per g of dry extract. All measurements were performed in triplicate.

The total anthraquinone content in the Near and Far Zone extracts of *B. macraei* was determined by colorimetric methods with aluminum chloride (AlCl_3_) [54,55]. For this, 500 mL of extract solution was added at a dry mass concentration of 1.0 mg mL^−1^ to 500 mL of a 2% solution of AlCl_3_ in ethanol. After 10 min, absorbance was measured at 485 nm using a UV–visible spectrophotometer (Rayleigh UV-2601, Beijing, China). All measurements were performed in triplicate. In addition, a blank solution was prepared, consisting of 500 mL of the extract with 500 mL of ethanol without AlCl_3_. A standard calibration curve was generated using known concentrations of emodin. Therefore, the total anthraquinone content was calculated and expressed in mg of emodin equivalents (EE) per g of dry extract. All measurements were performed in triplicate.

### 4.6. Determination of Antioxidant Activity of B. macraei Extracts

The antioxidant activity of the Near and Far Zone extracts of *B. macraei* was evaluated by three methodologies, stable free radical scavenging activity of 2,2′-diphenyl-1-picrylhydrazyl (DPPH∙), ferric-reducing power analysis (FRAP). and total reactivity antioxidant potential (TRAP), all of which are developed using UV–Visible spectrophotometric methods that seek to determine said activity in the extracts under study.

The scavenging activity of the stable free radical DPPH∙ was assessed by following the methodology described in the current literature [54,55,56]. The objective of this test was to evaluate the capacity of the extracts analyzed to eliminate the stable free radical DPPH∙. For this, 100 µL of each extract sample was added (in concentrations of 1.0 mg mL^−1^, 5.0 mg mL^−1^, and 10.0 mg mL^−1^) to 2.9 mL of DPPH solution at a concentration of 50 µM. It was then allowed to rest at room temperature for 15 min. Finally, the absorbance was determined at 517 nm by spectrophotometry against a blank sample, using the Rayleigh UV-2601 spectrophotometer. The absorbance was measured at zero minutes and after 15 min of rest with 100 µL of the sample and 2.9 mL of ethanol. All measurements were performed in triplicate. The experiments were standardized with gallic acid and Trolox^TM^ as a positive control. The scavenging activity will be determined by calculating the percentage of inhibition of the DPPH∙ radical, from which the IC_50_ value was obtained and reported.

The ferric-reducing power was determined following the methodology outlined in the current literature [54,55,57]. FRAP reagent was prepared by adding 300 mM Buffer solution, 10 mM 2,4,6-tri(2-pyridyl)-s-triacin (TPTZ), and 20 mM ferric chloride, at a ratio of 10:1:1, respectively. Briefly, 100 µL of the sample (which should be at 1 gr mL^−1^ concentration) was added to 3 mL of FRAP reagent and 300 µL of distilled water. The solution was mixed thoroughly and incubated for 30 min. The absorbance at 593 nm was measured spectrophotometrically using the Rayleigh UV-2601 spectrophotometer. Distilled water acted as a blank. A standard calibration curve was generated using known concentrations of Trolox^TM^ (0–120 mg mL^−1^). The reducing capacity of the extracts was expressed in mM TEAC. All measurements were replicated three times.

The TRAP trial followed the method described in previous studies [54,58]. The objective was to evaluate the antioxidant activity of secondary metabolites by spectrophotometric methods of the extracts prepared from the Far and Near Zone samples of *B. macraei*. For this, 150 µM of 2,2′-azino-bis (3-ethylbenzothialin-6-sulfonic acid) (ABTS) were added to 10 mM 2,2′-azo-bis(2-amidinopropane) (ABAP). It was mixed in a 1:1 ratio and incubated with constant shaking at 45 °C for 30 min, followed by cooling. The absorbance at 734 nm was measured spectrophotometrically using the Rayleigh UV-2601 spectrophotometer, every 10 s up to 50 s, against a blank solution corresponding to ABTS (the sample contained 1 mL of the mixture and 10 µL of the extract, which were at a ratio of 1:100). A standard calibration curve was generated using known concentrations of Trolox^TM^ (0–120 mg mL^−1^). The results are expressed in mM Trolox^TM^ equivalent antioxidant capacity (TEAC μM), using a Trolox^TM^ standard curve (0 to 120 mg L^−1^). All measurements were performed in triplicate.

### 4.7. GC-MS Technique: Identification of Volatile and Semi-Volatile Compounds from B. macraei Extracts

Volatile and semi-volatile compounds were identified by gas chromatography coupled to mass spectroscopy (GC-MS).

The extracts under study were analyzed by GC-MS (TFS, TRACE 1610, ISQ 7610 mass detector, Norwood, MA, USA) in order to identify the volatile and semi-volatile compounds present. The procedure can be found in the literature and is passed on to develop [54]. The extracts (1.0 L) were injected in mode (5 min) into an RTX-5MS column (30 m, 0.25 mm diameter, 5% diphenyl, and 95% dimethylpolysiloxane), with helium as the gas carrier at an instantaneous flow rate of 1.5 mL min^−1^ and a column pressure of 92.3 KPa. The injector temperature was 230 °C, generating the following thermal profile: the temperature was maintained for 3 min at 50 °C and then increased at a rate of 25 °C min^−1^ to 250 °C, before maintaining the temperature for 15 min.

The mass scan was operated in electron impact ionization (70 eV) mode, scanning from the m s^−1^ range of 30 to 600 in full scan mode. The mass spectra were compared with the internal spectral database. Compounds in the chromatograms were identified by comparing their mass spectra with those from the NIST/EPA/NIH mass spectral library [59].

Chromatographic peaks were considered “unknown” when their similarity index (MATCH) and inverse similarity index (RMATCH) were less than 850 and were discarded [60]. These parameters refer to the match ability of the target spectrum to the standard spectrum from the NIST MS Search 2.4 library (a value of 1000 indicates a best fit).

### 4.8. Isolation of Compounds from B. macraei Extracts

The leaf extracts of the Near and Far Zone species of *B. macraei*, prepared with ethyl acetate, were isolated by column chromatography and then characterized by nuclear magnetic resonance (^1^H and ^13^C NMR) and DEPT, HSQC, and HMBC analysis to identify and confirm the structures.

The extract was exposed to column chromatography on silica gel (SiliaFlash G60, 60–20 um, Quebec City, QC, Canada) for the stationary phase and the column was filled using a mixture of hexane and ethyl acetate for the mobile phase, initially containing 100% hexane, to subsequently increase the polarity gradient with ethyl acetate until it reached 100% of the latter. Then, the polarity percentage was increased by 1% to finally identify the isolated compound using a chromatographic plate, prior to identification tests by spectroscopy. Five fractions were isolated, each obtained from the ethyl acetate extract of the Far Zone and Near Zone samples, two of which were found to be a mixture of two compounds. One compound identified in the Far Zone matched compounds isolated from the Near Zone extract. Once the compounds were identified by nuclear magnetic resonance (NMR), the cytotoxic activity of each of them was evaluated.

### 4.9. Identification of Isolated Compounds from B. macraei Extracts

Compounds isolated from the Near and Far Zone extracts were identified by nuclear magnetic resonance (NMR). ^1^H, ^13^C (DEPT 135 and DEPT 90), HSQC, and 2D HMBC spectra were recorded in methanol deuterated solutions (MetOD) and referenced to residual peaks of *δ* 3.3 and 4.8 ppm and *δ* 49.2 ppm for ^1^H and ^13^C, respectively, on a Bruker Avance 400 digital NMR spectrometer, operating at 400.1 MHz for ^1^H and 100.6 MHz for ^13^C. Chemical shifts are expressed in *δ* ppm and coupling constants (J) in Hz. 

### 4.10. Cytotoxicity Activity of B. macraei Extracts and Isolated Phytoconstituents

To measure cytotoxic activity, a cell viability assay was conducted by staining cancer cells from the HT-29, PC-3, MCF7, and MCF10 lines with Sulforhodamine B (SRB). These lines were treated with increased phytoconstituent content and antioxidant activity from the extracts and isolated compounds of *B. macraei*. The Sulforhodamine B assay was performed according to established protocols in the current literature [54,61,62,63].

The extracts and isolated compounds of *B. macraei* should be kept protected from light and should be worked fresh. The mother solution should be kept at −20 °C and the working solution at 4 °C. Maintaining these conditions, they should be diluted in dimethylsulfoxide (DMSO) or ethanol to a concentration of 0.1 M, which will be the stock solution. With this, a concentration of 1 mM will be obtained, which will be the working concentration, from which four different concentrations were made (500 µg mL^−1^, 250 µg mL^−1^, 125 µg mL^−1^, and 62.5 µg mL^−1^). This should be followed by filtration.

The cell lines used should be washed and trypsinized before reaching 90–100% confluence. A homogeneous 1:1 solution (cell suspension: 0.4% (wt vol^−1^) trypan blue) should be achieved and the seeding cell density should be at 3 to 5 × 10^3^ cells/well. The cells are incubated at 37 °C, in an incubator with a 5% CO_2_ and 95% O_2_ atmosphere, and treated with the extracts prepared above for 72 h. The cells are then fixed with trichloroacetic acid for 1 h at 4 °C, washed by immersion, stained with 0.1% sulforhodamine B dissolved in acetic acid for 30 min, and washed 3 times, but this time with 1% acetic acid. Finally, the cell density is analyzed in a plate reader at 540 nm, and the viability of the cells under study will be estimated based on the percentage of inhibition. All measurements are in triplicate.

### 4.11. Statistical Analysis

Data are presented, using STATISTICA 7.0 software, as mean values ± standard deviation (SD). A distribution test was performed to identify the nature of the data obtained in the spectrophotometric tests (phytochemical profile and antioxidant activity). The results of this test for phenolic extracts, flavonoids, and anthraquinones were found to be parametric according to the Shapiro–Wilk normality test and were therefore subjected to the t-test for independent samples. As for the antioxidant tests, the normality test determined that only the DPPH test did not present a normal data distribution, which is why a T-test for independent samples was performed. The rest of the tests were found to be non-parametric, which is why the Kruskal–Wallis test (ANOVA of one factor) was performed. In all these cases, the confidence level is 95%. Additionally, to assess the correlation of environmental variables, a Pearson rank-order correlation matrix was conducted between phytoconstituents, antioxidant activity, in situ ETR, and the copper content of both the leaf and rhizosphere of the species.

## 5. Conclusions

The proximity of the industrial complex significantly impacts the increase in copper concentration in the rhizosphere, and this factor could lead to a decrease in the copper bioaccumulation capacity of *B. macraei*. This, in turn, results in an increased in situ electron transport rate related to its photosynthetic activity at a low concentration of copper and a reduction in the antioxidant activity of its extract as measured by DPPH and FRAP assays, but with no effect on its cytotoxic activity. However, at lower copper concentrations in the rhizosphere, even the photoinhibition mechanism would be affected and would favor antioxidant activity.

The antioxidant activity identified in the two species of *B. macraei* could be attributed to the presence of flavonoids such as Hispidulin, Cirsimaritine, and Isokaempferide, as well as the presence of oxygenated and non-oxygenated monoterpenes and sesquiterpenes identified in this study. The content of these compounds varies according to the proximity of the species to the industrial complex, which generates variations in antioxidant activity but not cytotoxic activity.

## Figures and Tables

**Figure 1 ijms-25-05993-f001:**
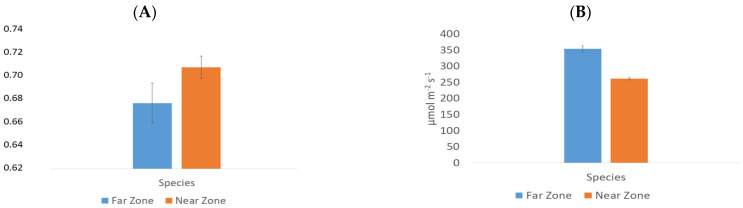
Photosynthetic physiology of the Near and Far Zone species of *B. macraei* (**A**) In situ photosynthetic capacity (Yield II). (**B**) In situ electron transport rate (ETR in situ, expressed in mmol electrons m^−2^ s^−1^). All the values showed significant differences with a *p* < 0.05. Irradiance in situ for the Near and Far Zones were 782 and 1108 µmol photons m^−2^ s^−1^, respectively.

**Figure 2 ijms-25-05993-f002:**
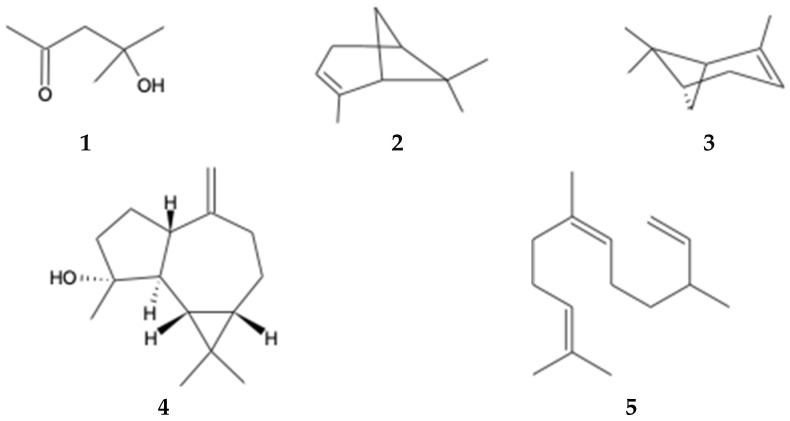
Compounds identified by GC-MS found in both study areas.

**Figure 3 ijms-25-05993-f003:**
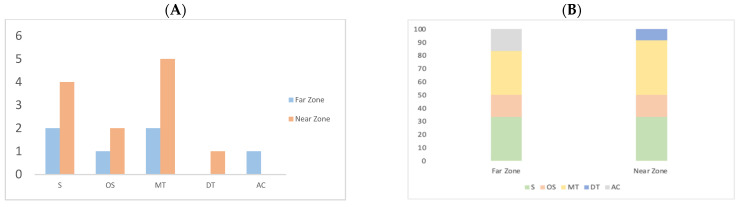
Quantity of compounds grouped by their chemical class: Sesquiterpene (S), Oxygenated sesquiterpene (OS), Monoterpene (MT), Diterpene (DT), and Aromatic compounds (AC). (**A**) Compound distribution (**B**) Percentage of compounds by chemical class.

**Figure 4 ijms-25-05993-f004:**
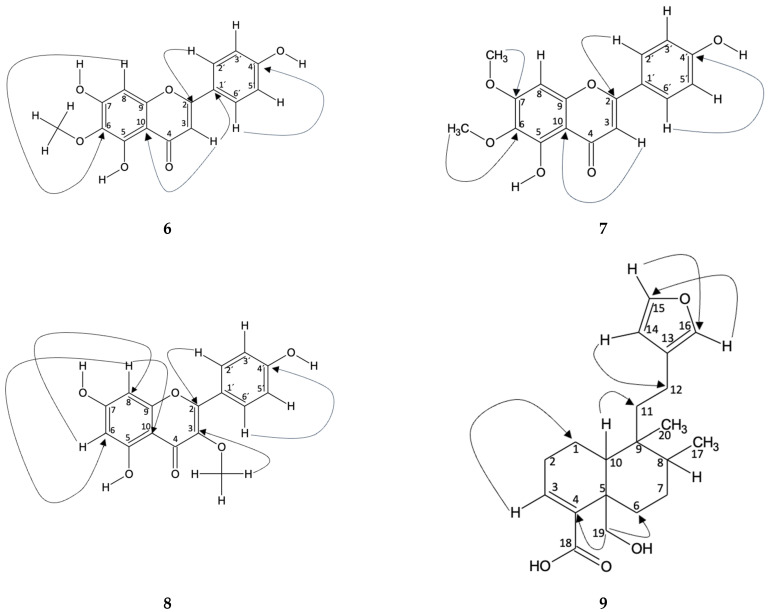
Correlation obtained from NMR spectroscopy analysis for the compounds **6**, **7**, **8** and **9** identified in Fractions 4, 9, 12, 13, and 14.

**Figure 5 ijms-25-05993-f005:**
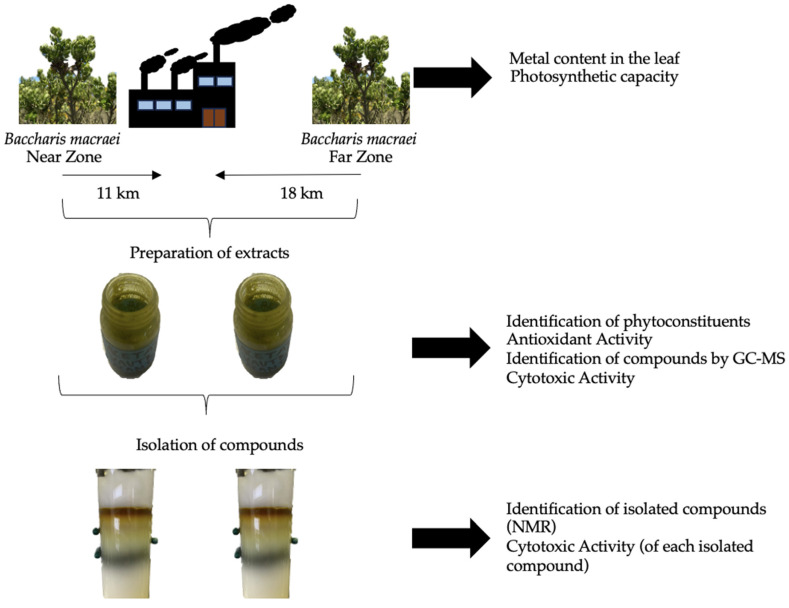
Experimental design.

**Figure 6 ijms-25-05993-f006:**
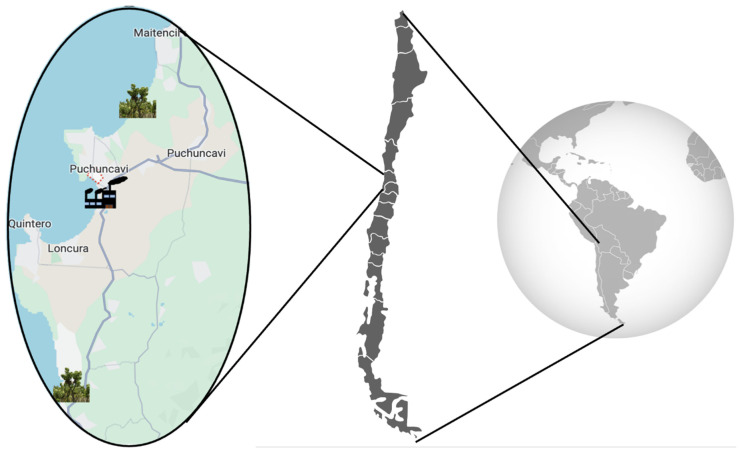
Sample distribution map in the industrial zone.

**Table 1 ijms-25-05993-t001:** Metal present in the rhizosphere and leaves of *B. macraei* and their concentrations, both in Near and Far Zone samples.

	Rhizosphere		Leaf	
Metal	Far Zone (mg kg^−1^)	Near Zone (mg kg^−1^)	Far Zone (mg kg^−1^)	Near Zone (mg kg^−1^)
Arsenic	6.27	6.065	<5	<5
Copper	29.45	44.8	15.1	11.4
Lead	<5	<5	<5	<5
Zinc	37.3	48.2	18.35	18.2

**Table 2 ijms-25-05993-t002:** Phytoconstituent content of ethyl acetate extracts prepared from the leaves of *B. macraei* of the contaminated and control species.

Phenols (mg L^−1^ GAE)		Flavonoids (mg L^−1^ QE)		Anthraquinones (mg L^−1^ EE)	
Far Zone	Near Zone	Far Zone	Near Zone	Far Zone	Near Zone
67.301 ± 2.956 ^a^	52.292 ± 1.089 ^b^	38.353 ± 0.744 ^a^	58.772 ± 1.190 ^b^	0.000 ± 0.803 ^a^	0.532 ± 0.572 ^a^

GAE: Gallic acid equivalents (phenols); QE: Quercetin equivalent (flavonoids) EE: Emodin equivalent (Anthraquinone). Superscript letters (a and b) of the same test represent significant differences, with *p* < 0.05. In cases where both test data have the same letter, they are considered to have no significant differences as *p* < 0.05.

**Table 3 ijms-25-05993-t003:** Antioxidant activity of ethyl acetate extracts prepared from *B. macraei* leaves compared with positive controls.

Organ/Positive Controls	DPPH (IC_50_)		FRAP (TEAC mM)		TRAP (TEAC mM)	
	Far Zone	Near Zone	Far Zone	Near Zone	Far Zone	Near Zone
Leaf	2.218 ± 0.015 ^a^	2.57 ± 0.116 ^a^	0.46 ± 0.001 ^a^	0.38 ± 0.004 ^b^	0.0366 ± 0.001 ^a^	0.0897 ± 0.002 ^b^
TROLOX^TM^	0.26 ± 0.00 ^b^		1.72 ± 0.02 ^c^		-	
GA	0.06 ± 0.00 ^c^		1.52 ± 0.07 ^d^		1.13 ± 0.01 ^c^	
BHA	-		-		1.06 ± 0.02 ^c^	

Positive controls: TROLOX^TM^, Gallic acid (GA), and Butylhydroxyanisole (BHA). Superscript letters (a–d) of the same test represent significant differences, with *p* < 0.05. In cases where both test data have the same letter, they are considered to have no significant differences as *p* < 0.05.

**Table 4 ijms-25-05993-t004:** Mass-coupled gas chromatography of *B. macraei* leaf extracts in ethyl acetate for the Near and Far Zone samples.

Peak Name	Far Zone	Near Zone	Chemical Class
	% of Rel. Area	% of Rel. Area	
2-Pentanone,4-hydroxy-4-methyl-	0.65	0.89	Other
Benzene, 1,3-dimethyl-	0.11	-	Aromatic compounds
α-pinene	0.33	1.63	Monoterpene
(-)-Spathulenol	0.81	1.84	Oxygenated sesquiterpene
γ-Terpinene	-	0.30	Monoterpene
Ylangene	-	1.64	Sesquiterpene
Cis-β-Farnesene	0.22	1.63	Sesquiterpene
γ-Amorphene	-	3.17	Sesquiterpene
Cadina-1(10),4-diene	-	5.72	Sesquiterpene
Neophytadiene	-	1.16	Diterpene
n-propyl acetate	0.14	-	Other
Methyl Isobutyl Ketone	0.22	-	Other
2-Propanol, 1-ethoxy-	0.24	-	Other
Glycinamide, N(2)-methyl-	0.95	-	Other
Acetic acid, butyl ester	0.09	-	Other
β-pinene	0.27	1.38	Monoterpene
Acetic acid, 2-ethylhexyl ester	0.23	-	Other
Copaene	0.44	-	Sesquiterpene
Ethyl hydrogen malonate	-	0.21	Other
Propanoic acid, ethyl ester	-	0.25	Other
β-Thujene	-	0.35	Monoterpene
(R)-1-Methyl-5-(1-methylvinyl) cyclohexene	-	0.66	Monoterpene
4(15),5,10(14)-Germacratrien-1-ol	-	0.42	Oxygenated sesquiterpene

**Table 5 ijms-25-05993-t005:** Cytotoxicity presented by ethyl acetate leaf extracts of *B. macraei* in tumorigenic and non-tumorigenic cell lines.

IC_50_ Values for Ethyl Acetate Extracts of *B. macraei*							
HT-29		PC-3		MCF-7		MCF-10	
Far Zone	Near Zone	Far Zone	Near Zone	Far Zone	Near Zone	Far Zone	Near Zone
>100	>100	>100	>100	>100	>100	>100	>100

**Table 6 ijms-25-05993-t006:** Cytotoxicity presented by the isolated compounds of *B. macraei* in tumorigenic and non-tumorigenic cell lines.

HT-29					PC-3					MCF-7					MCF-10				
4	9	12	13	14	4	9	12	13	14	4	9	12	13	14	4	9	12	13	14
>100	>100	>100	>100	>100	>100	>100	>100	>100	>100	>100	>100	>100	>100	>101	>100	>100	>100	>100	>100

**Table 7 ijms-25-05993-t007:** Pearson correlations for the phytoconstituent content, antioxidant activity in extracts, and photosynthetic physiology by ETR in situ and copper in the leaves and rhizospheres of the Far Zone.

	Phenols	Flavonoids	DPPH	FRAP	TRAP	Cu Leaf	Cu Rhizosphere	ETR In Situ
Phenols	-							
Flavonoids	0.762	-						
DPPH	−0.788	−0.202	-					
FRAP	−0.621	−0.981	0.007	-				
TRAP	−0.621	−0.981	0.007	1.000	-			
Cu leaf	−0.121	0.552	0.706	−0.703	−0.703	-		
Cu rhizosphere	−0.945	−0.931	0.545	0.842	0.842	−0.687	-	
ETR in situ	−0.638	−0.985	0.029	1.000	1.000	−0.209	0.854	-

**Table 8 ijms-25-05993-t008:** Pearson Correlations for the phytoconstituent content, antioxidant activity in extracts, and photosynthetic physiology by ETR in situ and copper in the leaves and rhizospheres of the Near Zone.

	Phenols	Flavonoids	DPPH	FRAP	TRAP	Cu Leaf	Cu Rhizosphere	ETR In Situ
Phenols	-							
Flavonoids	−0.064	-						
DPPH	0.879	−0.532	-					
FRAP	0.949	0.254	0.684	-				
TRAP	−0.924	0.441	−0.995	−0.756	-			
Cu leaf	0.316	−0.967	0.730	0.000	−0.655	-		
Cu rhizosphere	−0.316	0.967	−0.730	0.000	−0.982	−1.000	-	
ETR in situ	0.980	−0.264	0.957	0.866	0.655	0.500	−0.500	-

## Data Availability

The data presented in this study were obtained through laboratory analyses and they were not available in public databases.

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
