# Peer review of "Effect of the Proximity to the Quintero-Puchuncaví Industrial Zone on Compounds Isolated from *Baccharis macraei* Hook. & Arn: Their Antioxidant and Cytotoxic Activity"

_ijms, 2024, doi:10.3390/ijms25115993_

Round 1
Reviewer 1 Report
Comments and Suggestions for Authors
The manuscript "Effect of the proximity to the Quintero-Puchuncaví Industrial 2 zone on compounds isolated of Baccharis macraei Hook. & Arn 3 on their antioxidant and cytotoxic activity" describe relevant information and is appropriately presented. Minor revisions can be update before accepted for publication.
It is interesting to know the criteria that the authors used to discriminate the geographical proximity of the plant. It is recommended that the authors indicate why 11 km is a near zone?, and why 8 km is a far zone?
The list of references is exhaustive. Authors are recommended to review and choose those references with the greatest relevance, and discard references that do not provide sufficient relevance.
Author Response
Reviewer #1:
Manuscript Number: ID: ijms-3006900
Title: Effect of the proximity to the Quintero-Puchuncaví Industrial zone on compounds isolated of Baccharis macraei Hook. & Arn on their antioxidant and cytotoxic activity
Type of manuscript: Article
Journal name: IJMS
The authors are very grateful to the reviewers for their insightful comments. The manuscript has been accordingly modified, as much as possible. The manuscript is also attached with the changes made selected in colors depending on the reviewer (yellow or green). The changes are highlighted in yellow for reviewer #1
Response to specific comments:
It is interesting to know the criteria that the authors used to discriminate the geographical proximity of the plant. It is recommended that the authors indicate why 11 km is a near zone?, and why 8 km is a far zone?
It is important to highlight that, the distances at which the species under study are found are purely due to the availability of the species in the area (page 16). Therefore, this was added to the paragraph in Experimental design: “.... from two areas situated at different distances from the emission zone, according to the availability of the plant in the area”.
The list of references is exhaustive. Authors are recommended to review and choose those references with the greatest relevance, and discard references that do not provide sufficient relevance
The most relevant references have been reviewed and chosen, reducing those that were redundant.
Reviewer 2 Report
Comments and Suggestions for Authors
The authors of the manuscript "Effect of the proximity to the Quintero-Puchuncaví Industrial zone on compounds isolated of Baccharis macraei Hook. & Arn on their antioxidant and cytotoxic activity" described the biological and chemical properties of Baccharis macraei Hook growing in close and distant proximity to industrial areas associated with copper mining and processing. The parameters tested included antioxidant properties, identification of volatile compounds using GC-MS and NMR techniques, and chemopreventive properties. The manuscript has been prepared properly, but requires some corrections, which were suggested in the text of the attached file.

Author Response
Reviewer #2:
Manuscript Number: ID: ijms-3006900
Title: Effect of the proximity to the Quintero-Puchuncaví Industrial zone on compounds isolated of Baccharis macraei Hook. & Arn on their antioxidant and cytotoxic activity
Type of manuscript: Article
Journal name: IJMS
The authors are very grateful to the reviewers for their insightful comments. The manuscript has been accordingly modified, as much as possible. The manuscript is also attached with the changes made selected in colors depending on the reviewer (yellow or green). The changes are highlighted in green for reviewer #2.
Response to specific comments:
What did the authors understand by the rhizosphere?
The rhizosphere is the region of soil that surrounds plant roots and is influenced by them. This was added to the experimental design in parentheses after mentioning the rhizosphere “...from the rhizosphere (soil that surrounds plant roots in this study”.
There is no methodology for determining heavy metals. Please complete
Information was added on the methodology for measuring metals in the rhizosphere (page 16).
The extraction was carried out with what percentage of ethyl acetate?
The phytoconstituents of the extract were measured using technical level ethyl acetate.
It was completed in methodology (page 18).
Please specify which methods, although the methods used are listed later in the text
The method was added and therefore completed with the information: “Table 2 shows the phytoconstituent content of the ethyl acetate leaf extract of the species from both study areas, showing the concentrations of the phytoconstituents in studies obtained by UV-Visible spectrophotometry, which allows determining the concentration of the phytoconstituents. analytes in solution, the tests carried out are Phenols, Flavonoids and Anthraquinone”(page 4).
Please expand the abbreviations GAE, QE and EE in the table description
This information was added immediately below the table. GAE:Gallic Acid Equivalents (phenols); QE: Equivalent is Quercetin (flavonoids) EE: Equivalent is Emodin (Anthraquinone)
Describe the research methodology in more detail
By evaluating the cytotoxic activity of B. macraei extracts on the tumorigenic lines HT-29, PC-3 and MCF-7 and the non-tumorigenic line MCF-10, using the Sulforhodamine-B assay, which determines the concentration of cells that have survived treatment with the extracts under study, highlighting that no cytotoxic effect was observed in the tumorigenic cell lines HT-29, PC-3 and MCF-7, nor in the non-tumorigenic line MCF-10. (page 15).
Describe the full procedure for obtaining the extracts, how the evaporated extracts, how the evaporated extract was collected from the vacuum flask
This extraction methodology does not include the evaporation of the extract. Instead, this technique consists of extracting the solvent, specifically extracting the ethyl acetate, resulting in a dark green solid in both cases. This sentence was modified to avoid misunderstandings about the method: “The concentration of the macerate (extracts plus solvent) was carried out in a rotary evaporator (LabTech EV400H) “
How many repetitions were the tests carried out?
For the phytoconstituent methods, they were carried out in triplicate. This modification was added to the phytoconstituent estimation methodology.